# Multifractality without fine-tuning in a Floquet quasiperiodic chain

**Sthitadhi Roy[1*], Ivan M. Khaymovich[1], Arnab Das[2] and Roderich Moessner[1]**

**1** Max-Planck-Institut für Physik komplexer Systeme, Nöthnitzer Straße 38, 01187 Dresden, Germany
**2** Indian Association for the Cultivation of Science, Kolkata 700032, India

\* roy@pks.mpg.de

## Abstract

Periodically driven, or Floquet, disordered quantum systems have generated many unexpected discoveries of late, such as the anomalous Floquet Anderson insulator and the discrete time crystal. Here, we report the emergence of an entire band of multifractal wavefunctions in a periodically driven chain of non-interacting particles subject to spatially quasiperiodic disorder. Remarkably, this multifractality is robust in that it does not require any fine-tuning of the model parameters, which sets it apart from the known multifractality of *critical* wavefunctions. The multifractality arises as the periodic drive hybridises the localised and delocalised sectors of the undriven spectrum. We account for this phenomenon in a simple random matrix based theory. Finally, we discuss dynamical signatures of the multifractal states, which should betray their presence in cold atom experiments. Such a simple yet robust realisation of multifractality could advance this so far elusive phenomenon towards applications, such as the proposed disorder-induced enhancement of a superfluid transition.



# 1 Introduction

Multifractal wavefunctions are beautifully complex states, extended yet non-ergodic, comprising both rare high peaks and long polynomial tails of wavefunction amplitudes. The physics of multifractality is commonly associated with critical wavefunctions at Anderson localisation-delocalisation and quantum Hall plateau transitions [1–3]. Multifractality also appears in hierarchical and infinite-dimensional systems like random regular graphs, Bethe lattices, and more generally in fully connected random matrix ensembles and network models [4–12]. The presence of long-ranged physics diagnosed via correlation and localisation lengths unifies these two contexts. Hence, realising multifractality in an inherently short-ranged system, specifically systems with short-ranged hoppings and interactions unlike those for example, represented by power law banded or infinite ranged random matrices [5, 9], without fine-tuning to criticality poses not only an interesting and important theoretical challenge but is also desirable for a robust experimental realisation of multifractality and consequent applications.

We find that multifractality in a short-ranged system requires only relatively simple ingredients, namely a time-periodic modulation of a spatially-quasiperiodic system possessing a single particle mobility edge. Periodically driven systems, also known as, Floquet systems [13] have witnessed much interest recently with significant advances [14] in the understanding of their statistical mechanics [15–17] and phase structures [18] and in their experimental realisations with cold atoms [19]. Technically, the eigenfunctions of the Floquet unitary time-evolution operator over one period, $U$, encode the full information about the stroboscopic dynamics of the system, much like the eigenfunctions of the Hamiltonian of a static system [14]. At the same time, it has also been realised that single particle mobility edges occur naturally in simple incommensurate bichromatic potentials [20, 21]. At the level of one-dimensional lattice systems, this is related to the single particle mobility edges that generally exist in deformations of the Aubry-André model [22–28].

The central finding of this work is that, when the periodic drive hybridises the localised and delocalised states on either side of a mobility edge in a one-dimensional system with quasiperiodic potential, it gives rise to a *band of multifractal eigenstates* of the corresponding Floquet operator $U$. Remarkably, this multifractality exists in a finite range of parameters and thence requires no fine-tuning, while the states nonetheless show anomalous algebraic multifractal correlations similar–in some but not all–respects to the critical ones [29–33]. We present an effective random matrix Hamiltonian, which captures the numerically obtained multifractality remarkably well, bearing a family resemblance to the Rosenzweig-Porter random matrix ensemble [34], generalisations of which are known to host multifractal eigenstates [9,11,35–40].

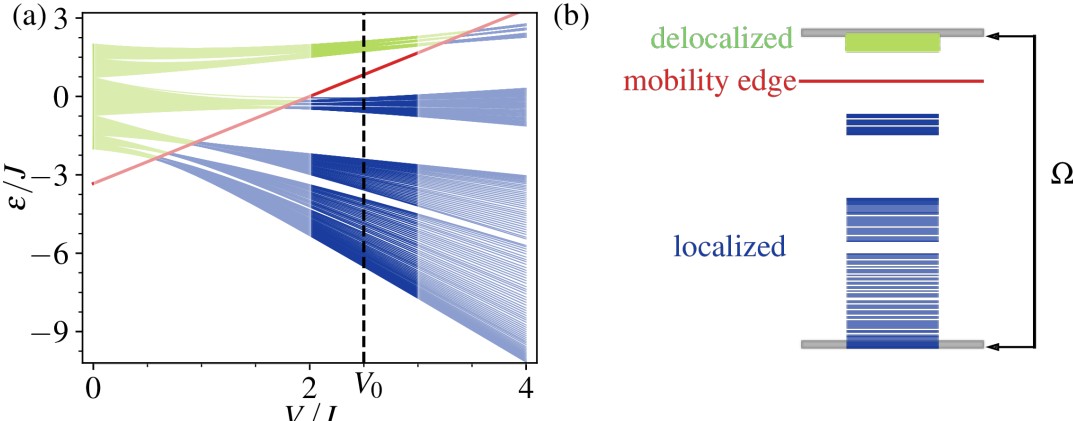

Figure 1: Schematic of the coupling between localised and delocalised states via the periodic drive. (a) The energy spectrum of the undriven Hamiltonian (1), where the colour shows the scaling of the inverse participation ratio with system size, with green corresponding to the delocalised states ($\sim L^{-1}$) and blue, localised states ($\sim L^0$). The red line denotes the mobility edge. (b) The energy levels corresponding to $V_0$ (denoted by the black dashed line in (a)) where a periodic drive with frequency $\Omega$ chosen to be slightly smaller than the bandwidth couples the delocalised and localised states approximately within the gray shaded windows.

## 2 Model and numerical results

Our starting point is a variant of the one-dimensional Aubry-André Hamiltonian

$$H = \sum_x \left[ J(\hat{c}_x^\dagger \hat{c}_{x+1} + \text{h.c.}) + V\nu(x)\hat{c}_x^\dagger \hat{c}_x \right] . \tag{1}$$

It comprises a simple nearest-neighbour hopping term alongside a potential

$$\nu(x) = \cos(2\pi\kappa x + \theta)/[1 - \mu\cos(2\pi\kappa x + \theta)], \tag{2}$$

quasiperiodic on account of its incommensurate wavevector, which we set to the golden mean, $\kappa = (\sqrt{5} + 1)/2$. The model exhibits a mobility edge [28] at an energy, $\varepsilon_{\text{ME}} = 2\,\text{sgn}(V)(|J| - |V|/2)/\mu$ as shown in Fig. 1(a). We set $J = 1$ and $\mu = -0.6$ throughout. Eigenstates with energies above and below $\varepsilon_{\text{ME}}$ are completely delocalised and exponentially localised, respectively. In all numerical analysis, we average the data over various values of the $\theta$ which is analogous to disorder averaging.

The system is driven by a time-periodic modulation of the amplitude of the quasiperiodic potential in the form of a square wave with frequency $\Omega$, mean $V_0$, and amplitude $\Delta V$. Such a protocol allows for the exact computation of the Floquet eigenstates, denoted henceforth as $|\phi\rangle$ via numerical diagonalisation of the $U$ which can be calculated relatively straightforwardly as $U = e^{-iH_+\pi/\Omega} e^{-iH_-\pi/\Omega}$ where $H_\pm$ denotes the Hamiltonian in the two steps of the square wave.

A common diagnostic for localisation properties of wavefunctions is their inverse participation ratio, $\text{IPR} = I_2 = \sum_x |\phi(x)|^4$ which scales with system size $L$ as $L^{-1}(L^0)$ for delocalised(localised) states in one dimension. The first signs of Floquet multifractality appear in the scalings of IPRs of the Floquet eigenstates. As $\Omega$ is chosen to be slightly smaller the bandwidth of the spectrum of the static Hamiltonian (1), with $V = V_0$ (see Fig. 1), the drive primarily couples states close to the top and bottom of the undriven spectrum, leaving largely unaffected all the localised and delocalised states in between. These latter two, together with our newly discovered multifractal states, are evident in Fig. 2(a)-(c), which now shows *three*

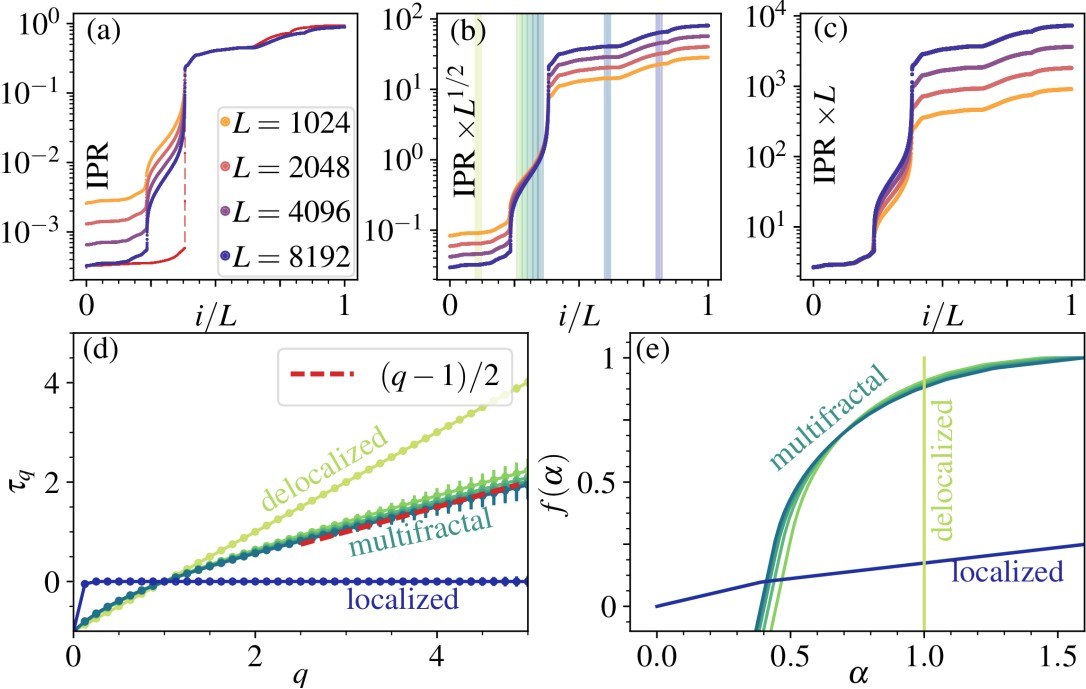

Figure 2: Characterisation of Floquet multifractal states. (a)-(c) The inverse participation ratio (IPR), shown for the Floquet eigenstates sorted in increasing order of $I_2$, for different system sizes $L$. The collapse of different segments of the data for different $L$, when the IPR is scaled with (a)$L^0$, (b)$L^{1/2}$, and (c)$L$ reflects the presence of localised, multifractal, delocalised states, respectively. The data in red in (a) shows the IPRs of static eigenstates (for $L = 8192$) for reference. (d) Averaging the moments over Floquet eigenstates in different windows highlighted by the vertical shaded regions in (b), $\tau_q$ is plotted as a function of $q$, where the colour corresponds to the window. While the localised (blue) and delocalised (yellow) states show the expected standard behaviour, the multifractal states (shades of green) have $\tau_q \approx D(q-1)$ at $q \gtrsim 1$ with $D \simeq 1/2$ (red dashed line). When averaged over all the multifractal states, the IPRs scale as $L^{-\tau_2}$, where $\tau_2 = 0.55 \pm 0.04$ close to $D \simeq 1/2$. The $\tau_q$s are extracted as the slope of a linear fit of $\log I_q$ versus $\log L$. Representative fits are shown in Fig. 3. (e) The corresponding spectrum of fractal dimensions, $f(\alpha)$ as function of $\alpha$ clearly shows the multifractal states distinct from both localised and delocalised cases. The system parameters are $V_0 = 2J$, $\Delta V = J/2$, and $\Omega = 2.74\pi J$, where the bandwidth of the undriven spectrum is $\approx 2.76\pi J$ (see Appendix B for effects of lower frequencies).

*distinct* scalings of the IPR. In disordered systems, since the energy spectrum varies across disorder realisations, labelling the Floquet eigenstates in increasing order of their IPRs as in Fig. 2 turns out to be rather convenient. However, we also study the quasienergy resolved IPRs by appropriately binning the data (see Appendix A).

A more complete characterisation of multifractality is via a generalised IPR and its scaling exponent $\tau_q$,

$$I_q(\phi) = \sum_{x=1}^{L} |\phi(x)|^{2q} \sim L^{-\tau_q}, \qquad (3)$$

where $D_q = \tau_q/(q-1)$ is known as the *fractal dimension*. For delocalised and localised states, $D_q = 1$ and $0$, respectively (for $q > 0$), whereas any other behaviour of $D_q$ implies multifractality. Multifractality is thus evidenced in $\tau_q$ shown in Fig. 2(d), where the localised and delocalised states show their standard behaviour. The multifractal states on the other hand seem

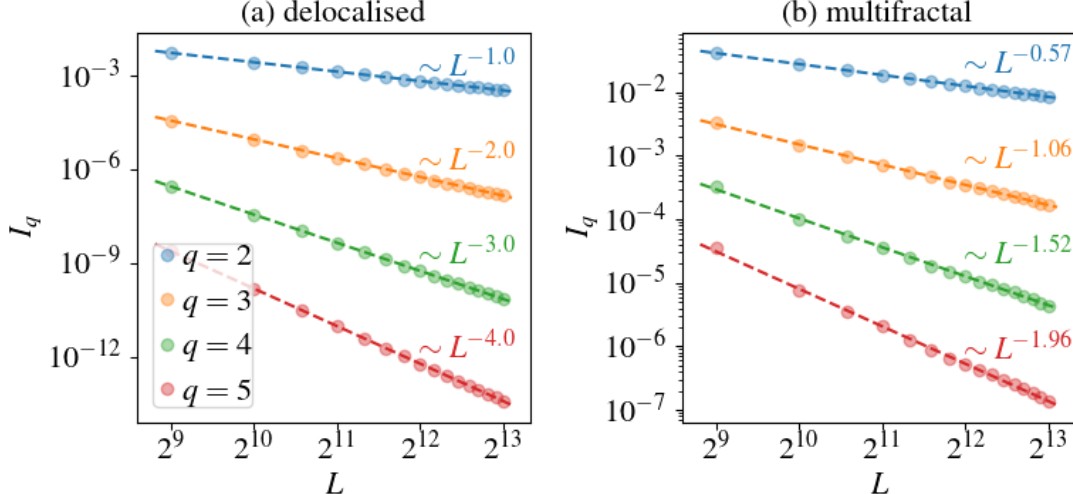

Figure 3: Extraction of $\tau_q$. Linear fit of $\log I_q$ versus $\log L$ for representative delocalised (a) and multifractal states (b). The circles show numerical data and the dashed lines show linear fits. The exponents mentioned in the plot suggest that $\tau_q = q - 1$ for the delocalised states and $\tau_q \approx D(q-1)$ with $D \simeq 1/2$ for the multifractal states.

to have show a good agreement with $\tau_q \approx D(q-1)$ with the $q$-independent $D_q = D \approx 1/2$, although one must notice that there is a spread in the behaviour of $\tau_q$ across the window of all multifractal states. When averaged over all the multifractal states, $D_q$ turns out to be $0.55 \pm 0.04$ for $q \gtrsim 1$.

To extract $\tau_q$ shown in Fig. 2(d), we do a linear fit of $\log I_q$ versus $\log L$ for all values of $q$ by selecting states in the shaded windows shown in Fig. 2(b). In Fig. 3, we show some examples of such fits for the delocalised and multifractal states for some representative values of $q$.

An equally fundamental measure of multifractality is the *spectrum of fractal dimensions*, $f(\alpha)$, which is defined via: the number of sites in a lattice system with total $L$ sites where the wavefunction intensity $|\phi(x)|^2 \sim L^{-\alpha}$ scales as $L^{f(\alpha)}$ [1]. $f(\alpha)$ is a rather powerful measure as it formally contains the information of all the $\tau_q$s via a Legendre transform, $f(\alpha) = q^*\alpha - \tau_{q^*}$ where $q^*$ is the solution of $\alpha = d\tau_q/dq$. $f(\alpha)$ for the Floquet multifractal states is shown in Fig. 2(e) which is strikingly distinct from that of a localised ($f(\alpha) = \lim_{\alpha_{\max}\to\infty} \alpha/\alpha_{\max}$) and delocalised ($f(\alpha) = 1$ for $\alpha = 1$ and $-\infty$ otherwise) state.

Turning to spatial correlations, multifractal wavefunctions exhibit algebraic behaviour concomitant with usual notions of criticality. To tease out multifractal behaviour, one again takes variable powers of the wavefunctions, to define the correlators $C(r; p, q) = \langle |\phi(x)|^{2p}|\phi(x+r)|^{2q}\rangle_{x,\text{disorder}}$, which are averaged both spatially and over disorder realisations. These then have a scaling form $\sim L^{-a_{p,q}} r^{-b_{p,q}}$, where $a_{p,q} = \tau_{p+q} + 1$ as shown in Fig. 4. This is similar to known critical multifractal wavefunctions but in our case $b_{p,q}$ is larger than the reported value of $\tau_p + \tau_q - \tau_{p+q} + 1$ [1, 30]. These states are thus genuinely fractal, not just mimicking fractality in their moments as in certain random-energy models [41].

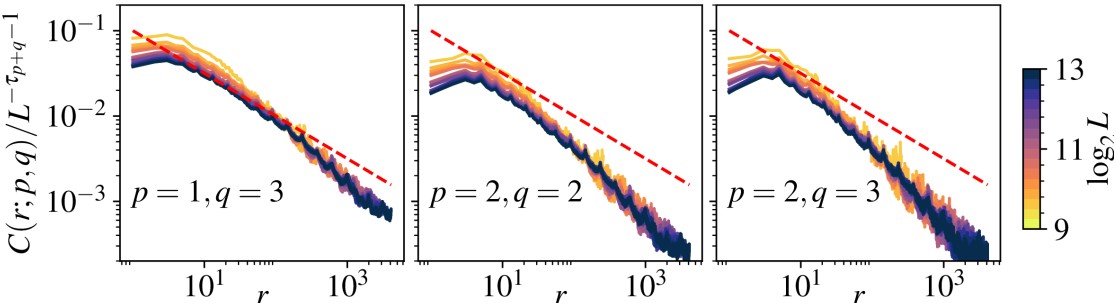

Figure 4: Power law decay of multifractal correlations. Multifractal spatial correlations $C(r; p, q)$ in space shown as a function of the distance $r$ for different values of $(p, q)$, where the collapse of the data suggests an algebraic scaling form $C(r; p, q) \sim L^{-\tau_{p+q}-1} r^{-b_{p,q}}$. The evident algebraic decay of the correlations is faster than for the previously studied critical multifractal states (red dashed lines).

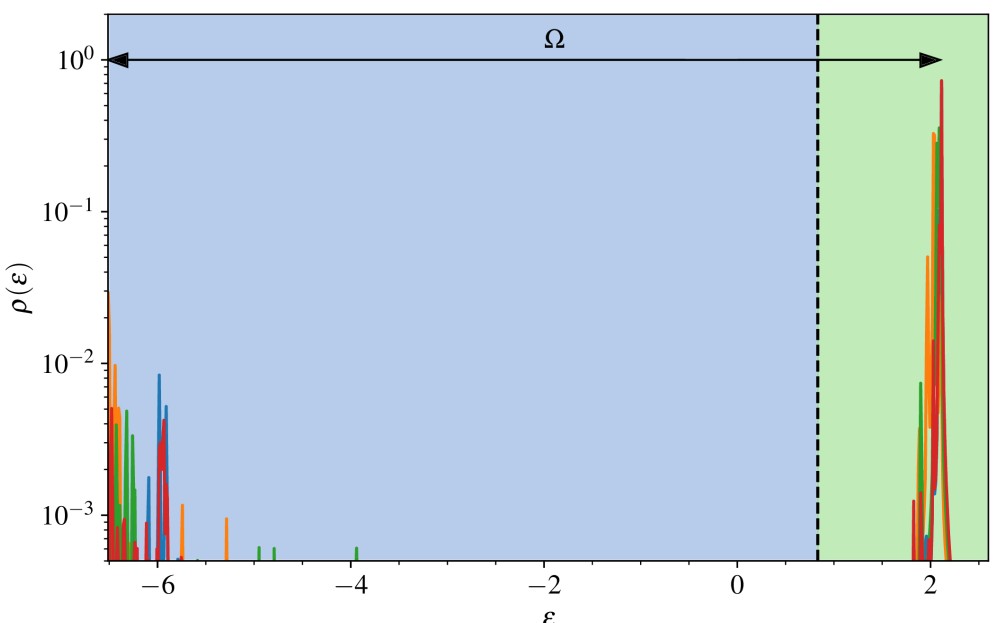

Figure 5: Spectral decomposition of the Floquet multifractal states. The spectral density of states $\rho(\varepsilon)$ for typical Floquet multifractal states chosen at random (different colors) show overwhelming contributions from the static eigenstates separated by $\varepsilon = \Omega$. The blue and green shaded regions correspond to the localized and delocalized part of the undriven spectrum, respectively. The black dashed line denotes the mobility edge, states near which do not get affected by the Floquet drive. The parameters correspond to Fig. 2.

## 3 Spectral decomposition of Floquet multifractal states

The underlying mechanism of Floquet generation of multifractality is the hybridisation of the localised and delocalised eigenstates of the undriven Hamiltonian close to the bottom and top of the spectrum, respectively. In this section, we provide evidence in form of the spectral decomposition of the Floquet eigenstates in terms of the those of the undriven Hamiltonian. To this order, we define the spectral density of states at energy $\varepsilon$ for a Floquet eigenstate $|\phi\rangle$

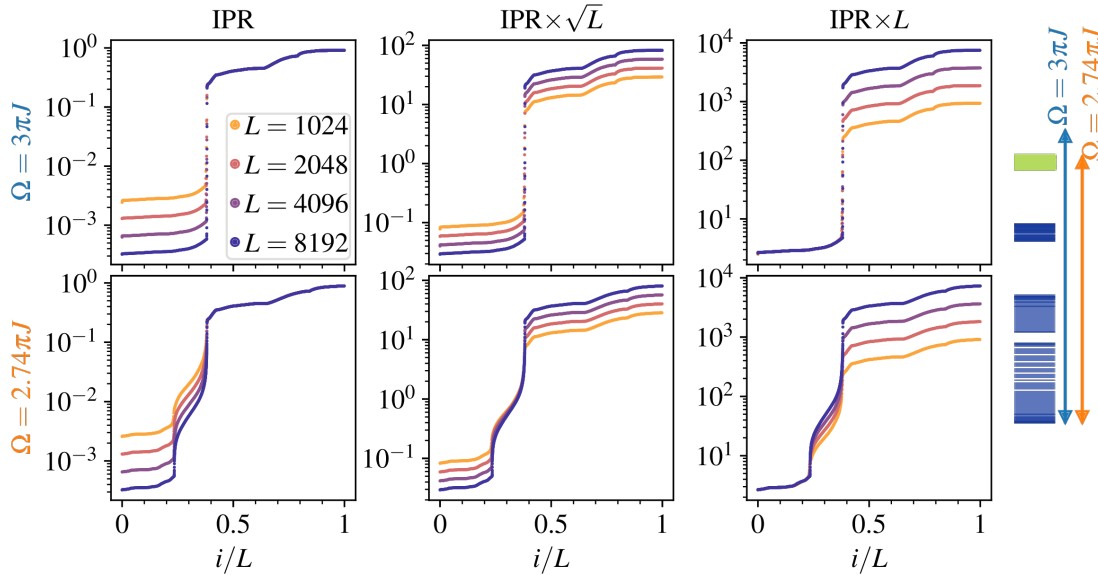

Figure 6: Dependence of multifractality on driving frequency. Absence and presence of multi-fractal states in the Floquet spectrum when the frequency of the driving is larger and smaller than the bandwidth of the undriven spectrum, respectively. The two rows correspond to the two values of $\Omega$, also shown schematically by the arrows next to the undriven spectrum, whereas the columns correspond to different scalings of IPR with $L$. The presence (absence) of multifractal states can be identified from the presence (absence) of finite fraction of states with IPR$\sim L^{-1/2}$ in the second column.

as

$$\rho_\phi(\varepsilon) = \frac{1}{\mathfrak{N}} \sum_{|\psi\rangle} |\langle\phi|\psi\rangle|^2 \frac{\eta}{\eta^2 + (\varepsilon - \varepsilon_\psi)^2}, \tag{4}$$

where $\varepsilon_\psi$ is an eigenvalue of the undriven Hamiltonian corresponding to the eigenstate $|\psi\rangle$, $\mathfrak{N}$ is a normalization factor to ensure $\sum_\epsilon \rho(\epsilon) = 1$ and $\eta$ is a small broadening factor. As expected, $\rho(\epsilon)$ for typical Floquet multifractal states chosen at random has overwhelming contributions from the localized and delocalized states near the bottom and top of the undriven spectrum as shown in Fig. 5.

To further corroborate this, we also show that the multifractal states appear only when the frequency of the driving, $\Omega$ is smaller than the bandwidth of the undriven spectrum. This is shown in Fig. 6 where the IPRs of the Floquet eigenstates show only two scalings $\sim L^0$ and $\sim L^{-1}$ when $\Omega$ is larger than the bandwidth. On the contrary as soon as $\Omega$ is tuned below the bandwidth, the multifractal states show their presence which can be identified by the IPR of the finite fraction of Floquet eigenstates scaling approximately as $L^{-1/2}$.

# 4 Effective random matrix model

We now turn towards understanding the Floquet multifractality within a random matrix framework. A central ingredient is the coupling between localised states $\{|l\rangle\}$, mediated by the delocalised states, $\{|d\rangle\}$, to which the localised ones couple through the driving. That the Floquet drive strongly couples eigenstates of the undriven Hamiltonian which are resonant (separated in energy by the frequency of the drive), is elegantly represented in the so called Shirley pic-

ture [42], where the time-periodic problem is mapped onto a static problem of hopping on a ladder, whose legs are copies of the chain and whose 'transverse' coupling is provided by the time-periodic drive. With our $\Omega$ just below the bandwidth, resonant coupling occurs between states close to the edges of the undriven spectrum on the localised and delocalised sides in the undriven spectrum. This can be modelled by a two-leg truncation of the Shirley ladder as couplings to higher legs come with an energy denominator of the order of the bandwidth and are therefore parametrically suppressed.

## 4.1 Derivation of effective random matrix Hamiltonian

We start with deriving the offdiagonal matrix elements of the effective random matrix Hamiltonian.

According to the Bloch-Floquet theorem, the eigenstates of a time periodic Hamiltonian $H(t) = H(t + 2\pi/\Omega)$ have a form $|\Phi(t)\rangle = |\phi(t)\rangle e^{-i\omega t}$, where $\omega$ is called the quasienergy and $|\phi(t)\rangle$ is itself periodic in time with frequency $\Omega$. Expressing $|\phi(t)\rangle$ in terms of its Fourier components $|\phi(t)\rangle = \sum_n |\phi_n\rangle e^{in\Omega t}$, the Schrödinger equation for $|\phi_n\rangle$ is given by

$$\omega|\phi_n\rangle = (H_0 + n\Omega)|\phi_n\rangle + \sum_{m\neq 0} H_m|\phi_{n-m}\rangle, \tag{5}$$

where the $H_m$ denote Fourier components of $H(t) = \sum_m H_m e^{im\Omega t}$. One can choose to work in the eigenbasis of $H_0$ denoted by $\{|\psi\rangle\}$ such that

$$|\phi_n\rangle = \sum_\psi c_{n,\psi}|\psi\rangle, \tag{6}$$

in which case, the Schrödinger equation can be recast as

$$\omega c_{n,\psi} = (\varepsilon_\psi + n\Omega)c_{n,\psi} + \sum_m \sum_{\psi'} c_{n-m,\psi'}\langle\psi|H_m|\psi'\rangle. \tag{7}$$

Since our driving frequency is only slightly smaller than the bandwidth, resonances can occur only between states near the top and the bottom of the static spectrum which are separated in energy by $\Omega$. Hence, we employ a simple two-leg Shirley ladder to analyze the system, as further legs correspond to processes involving multiples of $\Omega$, to which there are no corresponding resonances. Thus, we only keep $H_{\pm 1} \sim \Delta V \sum_x v_x \hat{n}_x$ and only the $n = -1$ and $n = 0$ sectors. Also, in our notation $|\psi\rangle = \sum_x \psi(x)c_x^\dagger|0\rangle$. Hence the matrix element $\langle\psi'|H_{\pm 1}|\psi\rangle = \Delta V \sum_x \psi'^*(x)\psi(x)v(x)$. These results can be put back in the equations for $c_{n,\psi}(t)$ as

$$\omega c_{-1,\psi} = (\varepsilon_\psi - \Omega)c_{-1,\psi} + \Delta V \sum_{\psi'}\sum_x c_{0,\psi'}\psi'^*(x)\psi(x)v(x),$$

$$\omega c_{0,\psi} = \varepsilon_\psi c_{0,\psi} + \Delta V \sum_{\psi'}\sum_x c_{-1,\psi'}\psi'^*(x)\psi(x)v(x). \tag{8}$$

We know that the multifractal states come from the hybridisation of the localised ($\{|l\rangle\}$) and delocalised states ($\{|d\rangle\}$) near the bottom and the top of the spectrum, respectively. Hence, in the two-leg Shirley ladder [42], the only states with relevant contributions are the delocalised ones from the $n = -1$ sector and the localized ones from the $n = 0$ sector. This is schematically shown in Fig. 7 where the participating undriven states are marked with a gray shaded window.

Hence the coefficients of interest are $c_{-1,d}$ and $c_{0,l}$. The equation for $c_{-1,d}$ reads

$$c_{-1,d} = \frac{\Delta V}{(\omega - \varepsilon_d + \Omega)} \sum_l \sum_x c_{0,l}\psi_l^*(x)\psi_d(x)v(x). \tag{9}$$

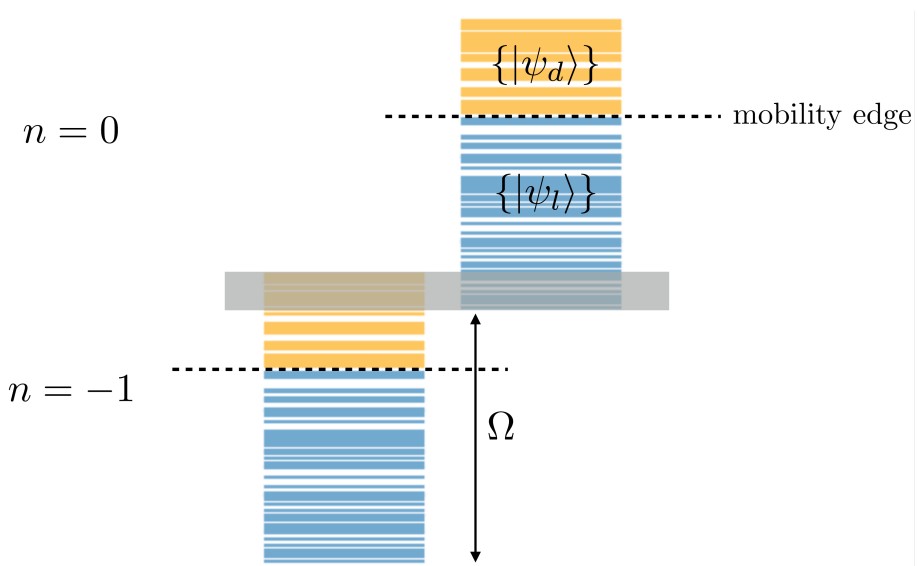

Figure 7: Schematic quasienergy spectrum of the undriven two-leg Shirley ladder. The gray shaded regions highlights the window of states which hybridize between the two legs when the driving is turned on.

Assuming that a localized state $|l_i\rangle$ is $\delta$-function localized at $x_i$, Eq. (9) can be plugged in Eq. (8) to obtain the equation for the coefficients of $c_{0,l_i}$ as

$$\omega c_{0,l_i} = \varepsilon_{l_i} c_{0,l_i} + \sum_{l_j} M_{l_i l_j} c_{0,l_j}, \tag{10}$$

where $M_{l_i l_j}$ denotes the off-diagonal matrix elements of the effective Hamiltonian,

$$M_{l_i l_j} = \sum_{d}{}' \frac{\psi_d(x_i) v(x_i) \psi_d^*(x_j) v(x_j) \Delta V^2}{(\omega - \varepsilon_d + \Omega)}. \tag{11}$$

which is the leading effective matrix element determining a resulting Floquet eigenstate at quasienergy $\omega$.

Here, a localised eigenstate of the undriven Hamiltonian $|l_i\rangle = |x_i\rangle$ is assumed to be $\delta$-function localised at $x = x_i$, and the primed sum denotes a sum over resonant delocalised states, highlighted by the gray shaded window in Fig. 1(b). This leads to a fully connected random matrix Hamiltonian within the localised states, with the undriven eigenenergies on the diagonal, and the $M_{l_i l_j}$ as the off-diagonal matrix elements. As an aside, we note that this model formally resembles the Rosenzweig-Porter random matrix ensemble, unlike which, however, it has a probability distribution of $M$ (Eq. (11)), denoted by $P(M)$ which is not Gaussian as we show in the following section 4.3.

The effective random matrix model allows us to connect the multifractality of the wave-functions to the statistical properties of the Floquet-generated matrix elements $M$ and their

scalings with system size.

## 4.2 Perturbative calculation of wavefunction intensity distributions from effective random matrix

Similar to the analysis in Ref. [9], we treat the corrections to the $\delta$-function localised eigenstates perturbatively $\phi_{l_i}(x_j) = M_{l_i l_j}/(\varepsilon_{l_i} - \varepsilon_{l_j})$. From the statistics of the perturbed wavefunctions, $f(\alpha)$ can be extracted as we will show now.

We derive the leading behavior of the distribution of wavefunction intensities $\mathscr{P}_{L|\phi|^2}$. Since, the offdiagonal terms of the effective random matrix Hamiltonian (11) typically decay with system size, they can be treated perturbatively as the thermodynamic limit is approached, in an analysis similar to Ref. [9]

The localized eigenstates of the unperturbed effective Hamiltonian are approximated to be $\delta$-function localized in space. As mentioned before, the localized state denoted by $|l_i\rangle$ is assumed to be localized at $x_i$. To leading order in $M$, the wavefunction intensity at any site can then be written as

$$|\phi_{l_i}|^2(x_j) = \delta_{x_i, x_j} + \frac{M_{l_i l_j}^2}{(\varepsilon_{l_i} - \varepsilon_{l_j})^2}. \tag{12}$$

Since we are interested in the probability distribution, $\mathscr{P}_{L|\phi|^2}$, we consider its generating function

$$G(s) = \langle e^{isL\phi^2} \rangle \Rightarrow (-i)^q \partial_t^q G(s)|_{s=0} = \langle (L\phi^2)^q \rangle, \tag{13}$$

where $\langle \cdot \rangle$ denotes the average over sites and disorder realizations. The regular part of the generating function $G(s)$ coming from the perturbative couplings is given by

$$G_{\text{reg}}(s) = \langle e^{isL\phi^2} \rangle = \int d(L|\phi|^2) \, \mathscr{P}_{L|\phi|^2} e^{isL|\phi|_{reg}^2}. \tag{14}$$

For simplicity, we consider the energy differences $(\varepsilon_l - \varepsilon_{l'})^2 = \Delta_{l,l'}^2$ belonging to a Gaussian distribution $P_\Delta = e^{-\Delta^2/2\sigma_\Delta^2}/\sqrt{2\pi\sigma_\Delta^2}$, the width of which is assumed to have no scaling with $L$. With these assumptions, the regular part of $G(s)$ can be expressed as

$$
\begin{aligned}
G_{\text{reg}}(s) &= \int_{-\infty}^{\infty} d\Delta \, P_\Delta \int_{-\infty}^{\infty} dM \, P_M \, e^{isLM^2/\Delta^2} \\
&= \int_{-\infty}^{\infty} dM \, P_M \, e^{-\sqrt{-2isLM^2}/\sigma_\Delta}.
\end{aligned}
\tag{15}
$$

Our quantity of interest, $\mathscr{P}_{L|\phi|^2}$, is the inverse Fourier transform of $G_{\text{reg}}(s)$,

$$\mathscr{P}_{L|\phi|^2} = \int_{-\infty}^{\infty} ds \, e^{-isL|\psi|^2} \int_{-\infty}^{\infty} dM \, P_M \, e^{-|M|\sqrt{-2isL}/\sigma_\Delta}. \tag{16}$$

In the thermodynamic limit, in the integral over $s$ in Eq. (16), the range of $s$ which contributes is such that $sL|\psi|^2 \sim 1$. This implies that, if we are interested in the probability of the wavefunction intensity scaling as $L^{-\alpha}$, we must consider $s \sim L^{\alpha-1}$. Assuming a single-parameter scaling of the distribution $P_M = P(m = M/M_0) \times M_0$ with the scaling $M_0 \sim L^{-\gamma/2}$, and the convergence of the integral in Eq. (15) as

$$G_{\text{reg}}(s) = \int_{-\infty}^{\infty} dM \, P_M \, e^{-\sqrt{-2isLM^2}/\sigma_\Delta} \equiv g(M_0\sqrt{-2isLM^2}/\sigma_\Delta), \tag{17}$$

one can expand the latter in a series and truncate at the first order for $\alpha < \gamma$

$$G_{\text{reg}}(s) = 1 - c\sqrt{-2isL^{1-\gamma}}/\sigma_{\Delta}, \tag{18}$$

with an $L$-independent constant $c$. We will discuss the question of the convergence of the above mentioned series and scaling of the distribution function $P_M$ for the next section. The leading behavior in $\mathscr{P}_{L|\phi|^2}$ is

$$\mathscr{P}_{L|\phi|^2} \sim \frac{L^{(1-\gamma)/2}}{(L|\phi|^2)^{3/2}} \int_{-\infty}^{\infty} d(sL|\phi|^2) \left[ e^{-isL|\phi|^2} (sL|\phi|^2)^{1/2} \frac{c(-2i)^{1/2}}{\sigma_{\Delta}} \right], \tag{19}$$

which in terms of $\alpha$ can be expressed as

$$\mathscr{P}_{L|\phi|^2} \sim C_1 \frac{L^{(1-\gamma)/2}}{(L|\phi|^2)^{3/2}} \sim C_1 \frac{L^{(1-\gamma)/2}}{(L^{1-\alpha})^{3/2}}, \tag{20}$$

where $C_1$ depends at most logarithmically with $L$. The normalization of the probability distribution and the wavefunction intensities put further bounds on $\alpha$.

- Normalization of the distribution $\int_0^{\infty} d(L|\phi|^2) \mathscr{P}_{L|\phi|^2} = 1$ implies that the lower bound of $L|\phi|^2$ and hence the upper bound of $\alpha$ is given by $L^{(1-\gamma)/2}L^{(\alpha_{max}-1)/2} \sim L^0$, which implies $\alpha_{max} = \gamma$ .

- Normalization of the wavefunction $\int_0^{\infty} d(L|\phi|^2) L|\phi|^2 \mathscr{P}_{L|\phi|^2} = 1$ implies that the upper bound of $L|\phi|^2$ and hence the lower bound of $\alpha$ is given by $L^{(1-\gamma)/2}L^{(-\alpha_{min}+1)/2} \sim L^0$, which implies $\alpha_{min} = 2 - \gamma$ .

Within these bounds of $\alpha$, the spectrum of fractal dimensions, $f(\alpha)$ can be calculated as follows. Since $\mathscr{P}_{L|\phi|^2} d(L|\phi|^2) = P(\alpha)d\alpha$, one obtains up to logarithmic corrections

$$P(\alpha) = L|\phi|^2 \mathscr{P}_{L|\phi|^2} = L^{f(\alpha)-1} \tag{21}$$

which in turn yields Eq. (5):

$$f(\alpha) = 1 + (\alpha - \gamma)/2; \quad 2 - \gamma < \alpha < \gamma. \tag{22}$$

## 4.3 Scaling of distribution of off-diagonal elements of random matrix Hamiltonian

In order to justify the assumptions in Sec. 4.2, we analyse the distribution $P_M$ in detail. We focus on the distribution of the absolute value $|M|$ as the probability distribution of $M_{l_i l_j}$ is symmetric with respect to the sign of the matrix element. Constructing the probability distribution $P(M)$ numerically, Fig. 8(a), shows a strongly non-Gaussian distribution with polynomial tails consistent with the Levy random matrices [37, 43].

Rescaling the distribution as

$$P(M/M_0 = m) = P(M = M_0 \cdot m)M_0 , \tag{23}$$

with $M_0 = L^{-\nu}$ gives a reasonable collapse for $\nu = 0.9 \pm 0.2$ in the considered range of system sizes $L = 2^9 - 2^{13}$, (see Fig. 8(b)). $P(M/M_0 = m) \sim m^{-2a}$ decays polynomially with $m = M/M_0$ saturating at $m \simeq m_0 \sim 1$. The best fit

$$P_{fit}(M/M_0 = m) \sim \frac{1}{m^b(m^2 + m_0^2)^c} , \tag{24}$$

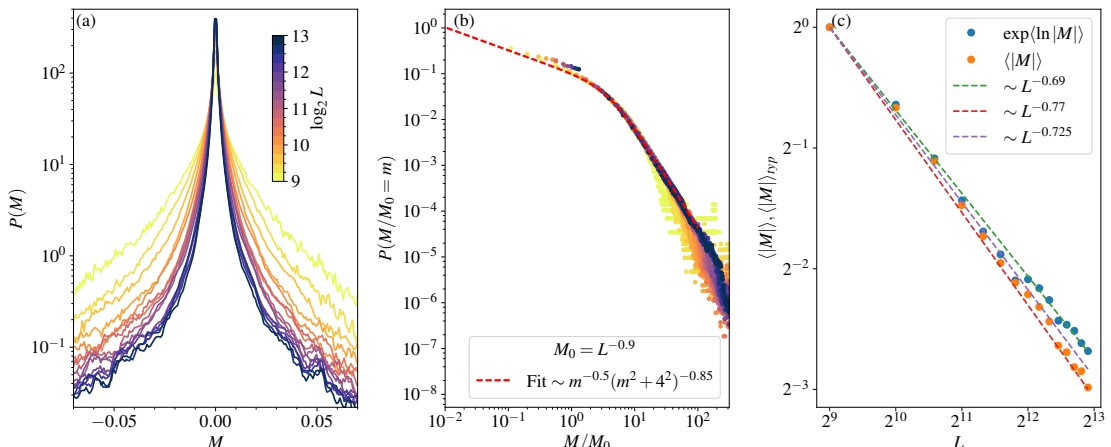

Figure 8: Probability distribution of $M$. (a) $P(M)$ calculated from the distribution of the elements in Eq. (11) for different $L$ (denoted by the color scale) is *not* Gaussian. (b) The collapsed probability distribution $P(M/M_0)$ with the corresponding polynomial fit (see legend for details). (c) The scaling of $\langle |M| \rangle_P$ (mean) and $\exp[\langle \ln M \rangle_P]$ (typical) with $L$. Dashed lines of the corresponding colors show fits with algebraic decay with exponents $\approx 0.78$ and $\approx 0.69$, respectively. The algebraic decay $\sim L^{-(2-D)}$ with $D = 0.55 \pm 0.04$ expected from multifractal analysis is shown by a black dashed line. Parameters are the same as in Fig. 2.

gives $b \simeq 0.5$, $m_0 \simeq 4$, $c = a - b/2$, and $a \simeq 1.1$ for $\nu = 0.9$ (see Fig. 8(b)). This confirms the first assumption of Sec. 4.2. However, the accuracy of the extracted parameter $\nu$ which assumed to be equal to $\gamma/2$ is of order of 20 %.

The integral (15) of the form

$$g(AM_0) = \int_{-\infty}^{\infty} dM \, P_M \, e^{-A|M|} = \int_{-\infty}^{\infty} dm \, P(m) \, e^{-AM_0|m|}, \qquad (25)$$

with $\mathrm{Re}[A] > 0$ converges both at $m = 0$ (as $b < 1$) and at $m \to \infty$ (due to exponential decay of the integrand). The first-order expansion (18) in $AM_0$ is valid for the parameters $a > 1$ corresponding to the converging first moment $\langle |M| \rangle$ in the limit $AM_0 \to 0$. As the best fit gives $a \simeq 1.1$ this justifies the second and the final assumption of Sec. 4.2.

For a more accurate estimate of $\nu$, we calculate the mean $\langle |M| \rangle$ and the typical $\langle |M| \rangle_{typ} \equiv \exp\langle \ln |M| \rangle$ values of the distribution as both of them should be governed by $M_0$. The numerical calculations show the algebraic decay of both mean and typical with the exponents $\nu_{mean} = 0.78$ and $\nu_{typ} = 0.69$ rather close to the expected value $\nu = \gamma/2 = 1 - D/2 \simeq 0.725$ (see Fig. 8(c)).

The difference between $\nu_{mean}$ and $\nu_{typ}$ should be considered as an error bar estimate as the points for different $L$ scattered within this interval. As mentioned previously, since $\tau_q$ and $f(\alpha)$ are related via Legendre transformation one obtains from this analysis and Eq. (22), $\tau_q = (2 - \gamma)(q - 1)$ with $2 - \gamma = 0.53 \pm 0.11$. This is in rather close agreement with the result numerically obtained in Fig. 2(d) thus validating the random matrix model. Thus, the scaling analysis of the distribution $P_M$ confirms both assumptions of Sec. 4.2.

The underlying origin of multifractality is hence the non-trivial mixing of localised states mediated by the delocalised states as an effect of the Floquet drive, thus linking our fully short-range model to an effective long-ranged random matrix ensemble.

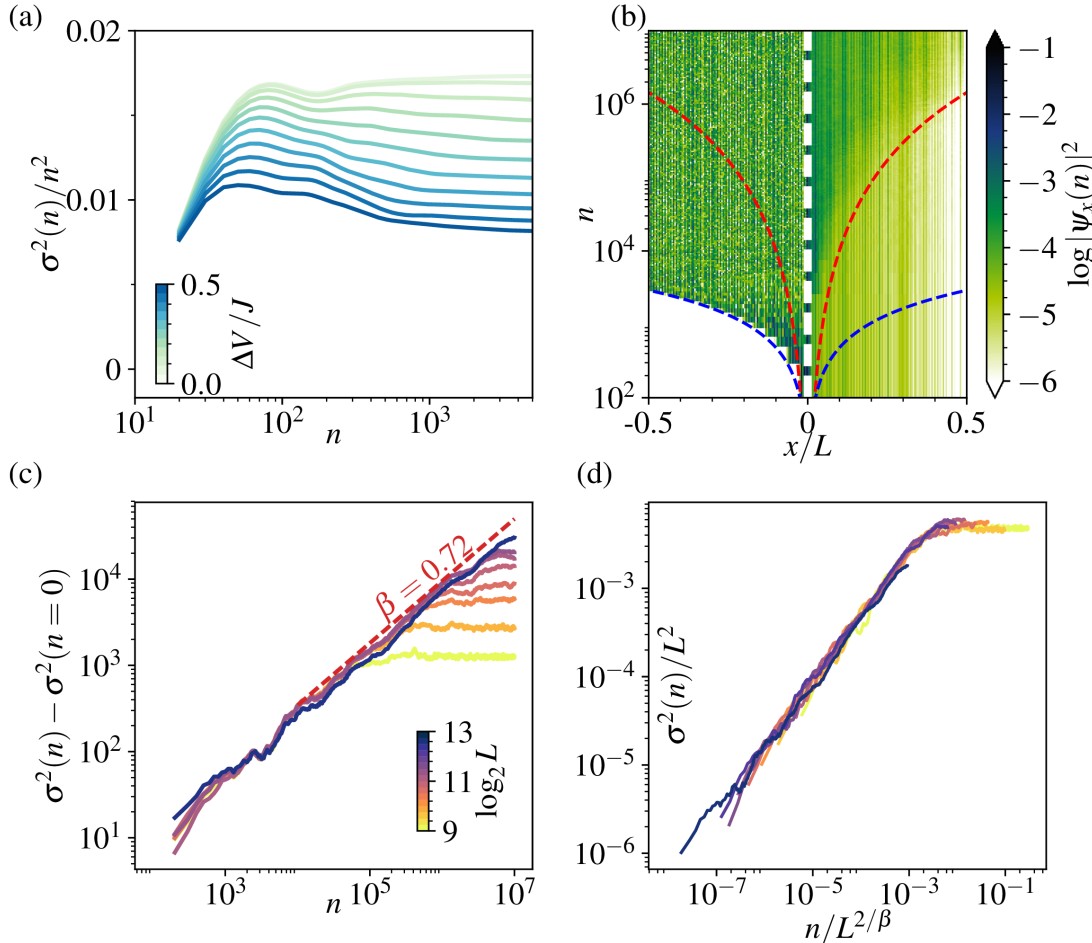

Figure 9: Signatures of multifractality in wavepacket spreading. (a) Ballistic spreading, $\sigma^2(n)/n^2$, is suppressed as increasing the drive $\Delta V$ converts delocalised into multifractal states. (b) Local density $|\psi_n(x)|^2$ as a colour map over $x$ and $n$, where $x < (>)0$ shows the ballistic (subdiffusive) dynamics due to delocalised (multifractal) states. The blue (red) dashed lines corresponding to ballistic (subdiffusive) dynamics indicate the difference between the two. The initial state is localised at the origin for $x < 0$, while for $x > 0$, all the delocalised components have been projected out of this state. (c) Subdiffusive behaviour in $\sigma^2(n)$ of multifractal states with exponent $\beta \approx 0.72$. (d) Collapse of the data shown in (c) as $\sigma^2 \sim L^2 \mathscr{F}(n/L^{2/\beta})$. For (a)-(b), $L = 4096$, and rest of the parameters are the same as that of Fig. 2.

## 5   Wavepacket dynamics

How can this physics be probed experimentally? An auspicious setting is provided in cold atom experiments, where incommensurate potentials in one-dimension have already been realised, for instance by superposing optical lattices of wavelengths 532 nm and 738 nm [44] and periodic drives have recently been prominently investigated by sinusoidally modulating laser intensities [19]. As this naturally permits dynamical measurements, we address a conceptually simple process, the spreading of an initially localised wavepacket, $|\psi_0\rangle$, focusing on signatures of multifractality. Spreading is conveniently quantified by $\sigma(n)$ via

$$\sigma^2(n) = \langle \psi_n | \hat{x}^2 | \psi_n \rangle - \langle \psi_n | \hat{x} | \psi_n \rangle^2, \tag{26}$$

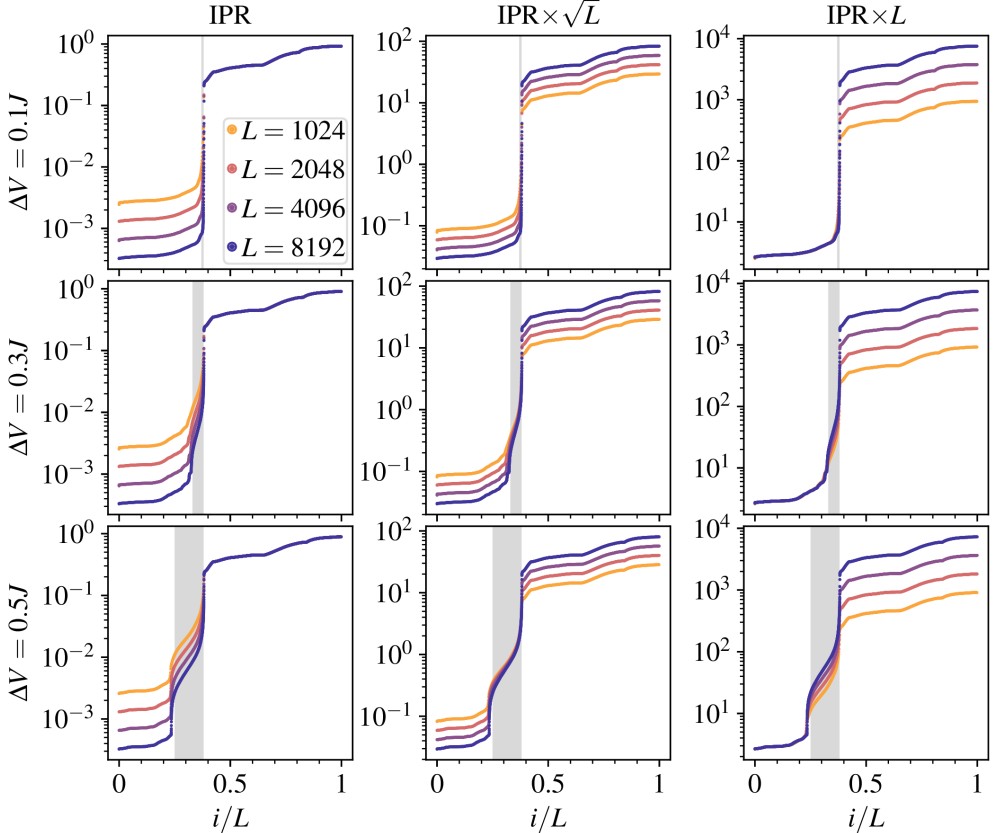

Figure 10: Fraction of multifractal states in the Floquet quasienergy spectrum for different $\Delta V$. While the three rows of plots correspond to $\Delta V = 0.1J$, $0.3J$, and $0.5J$, respectively, the three columns correspond to the three different scalings of the IPR. The multifractal states denoted by the gray shaded window are identified by noting the collapse of the IPR$\times\sqrt{L}$ for different $L$, and their fraction grows with $\Delta V$.

averaged over disorder, where $|\psi_n\rangle = U^n|\psi_0\rangle$ is the wavepacket after $n$ driving periods. The presence of an extensive number of delocalised states in the eigenbasis of $U$ leads to a ballistic leading behaviour $\sigma^2(n) \sim n^2$.

In the presence of multifractal states, subleading behaviour emerges as $\sigma^2(n) \sim \lambda_1 n^2 + \lambda_2 n^\beta$. This becomes increasingly visible with a growing $\Delta V$ which, as we observe from our numerics, increases the fraction of multifractal states. This is visible in the plot of $\sigma^2(n)/n^2$, Fig. 9(a), where the amplitude of the ballistic growth is continuously suppressed with increasing $\Delta V$. As a matter of principle, to accurately capture $\beta$, it is desirable to remove the dominant contribution of the delocalised states, which can be achieved in theory by removing the projection of the initial wavefunction onto them.

The (normalised) projected initial state $|\tilde{\psi}_0\rangle$, now has the leading contribution to the spreading from the multifractal states. It shows a much slower growth of $\sigma^2$ as shown in Fig. 9(b). The dynamics is in fact *subdiffusive* with $\beta \approx 0.72$, Fig. 9(c). A collapse of the data suggests a scaling form $\sigma^2(n) \sim L^2 \mathscr{F}(n/L^{2/\beta})$ where $\mathscr{F}(x) \sim x^\beta$ in the scaling regime and $\mathscr{F}(x) \sim 1$ as $x \to \infty$. This is not unlike the results obtained on hierarchical lattices [45].

The fact that the subleading behavior in the wavepacket spreading due to the multifractal states becomes stronger with increasing $\Delta V$ is a consequence of the fact that the fraction of multifractal states in the spectrum of $U$ increases with increasing $\Delta V$. We confirm this by providing the scaling of the Floquet eigenstates for different $\Delta V$, see Fig. 10.

# 6 Unequal time density correlators and wavefunction moments

Alternatively, we note that the time-averaged unequal time density correlators can reproduce all moments of the eigenstate wavefunctions and hence the full multifractal spectrum as we show in this section.

Since, the initial state is taken to be $\delta$-function localized at $x_0$, $|\psi_0\rangle = |x_0\rangle$, the quantity of interest is the $n$-time density correlator measured at the site $x_0$,

$$\mathcal{R}(x_0; t_1, \cdots, t_n) = \langle x_0 | \left( \prod_{i=1}^{n} \hat{n}_{x_0}(t_i) \right) | x_0 \rangle = \sum_{\{\phi_i\}} \left( \prod_{i=0}^{n} |\phi_i(x_0)|^2 \right) \left( \prod_{i=1}^{n} e^{i(E_{\phi_{i-1}} - E_{\phi_i}) t_i} \right), \quad (27)$$

where we use $\langle \phi_j | \hat{n}_{x_0} | \phi_i \rangle = \phi_j^*(x_0) \phi_i(x_0)$. The infinite time average of $\mathcal{R}$ is related to the moments of the eigenstates averaged over the spectrum as follows,

$$\lim_{t \to \infty} \left( \prod_{i=1}^{n} \int_0^t \frac{dt_i}{t} \right) \mathcal{R}(x_0; t_1, \cdots, t_n) = \sum_{\{\phi_i\}} \left( \prod_{i=0}^{n} |\phi_i(x_0)|^2 \right) \left( \prod_{i=1}^{n} \delta_{\phi_{i-1}, \phi_i} \right)$$

$$\Rightarrow \mathcal{R}_\infty^{(n)}(x_0) = \sum_\phi |\phi(x_0)|^{2(n+1)}. \quad (28)$$

In the next step, we take an average over the initial conditions which essentially gives

$$\mathcal{R}_\infty^{(n)} = \frac{1}{L} \sum_{x_0=1}^{L} \mathcal{R}_\infty^{(n)}(x_0) = \frac{1}{L} \sum_{\phi=1}^{L} \sum_{x_0=1}^{L} |\phi(x_0)|^{2(n+1)}. \quad (29)$$

So Eq. (29) implies that $\mathcal{R}_\infty^{(n)}$ is the $2(n+1)^{\text{th}}$ moment of eigenstates averaged over all the eigenstates. For instance $n = 0$ gives just the normalization, $n = 1$ gives the IPR, and so on.

The moments of the eigenstates, averaged over the eigenstates, can thus be useful because they can carry non-trivial information about the presence of multifractal states in the spectrum. For example, consider that the Floquet spectrum has $f_1 L$ localized states, $f_2 L$ delocalized states, and $f_3 L$ multifractal states where $f_i$s denote the respective fractions and $f_1 + f_2 + f_3 = 1$. In such a scenario, let us consider $\mathcal{R}_\infty^{(1)}$ which consists of the information of the IPRs of the Floquet eigenstates. The numerical results presented in Fig. 2 suggest that the IPRs of the multifractal states approximately scale as $L^{-\tau_2}$, where $\tau_2 = 0.55 \pm 0.04$. Hence, $\mathcal{R}_\infty^{(1)}$ is expected to have an approximate form

$$\mathcal{R}_\infty^{(1)} \sim \frac{1}{L} \sum_{\alpha=1}^{L} \text{IPR}_\alpha$$

$$\sim f_1 + \frac{f_2}{L} + \frac{f_3}{L^{\tau_2}}. \quad (30)$$

So by extrapolating the data as function of $1/L$ to zero, one can obtain the $L \to \infty$ value which can be subtracted, and the leading behavior with $L$ can be obtained. As shown in Fig. 11 the procedure yields a slope of $-0.54$, which is indeed within the error bars of $\tau_2$. A similar analysis can be done for the higher moments.

# 7 Conclusion

In conclusion, periodically driving a system with a single particle mobility edge can yield robust multifractal states. Note that a single particle mobility edge exists rather generically for

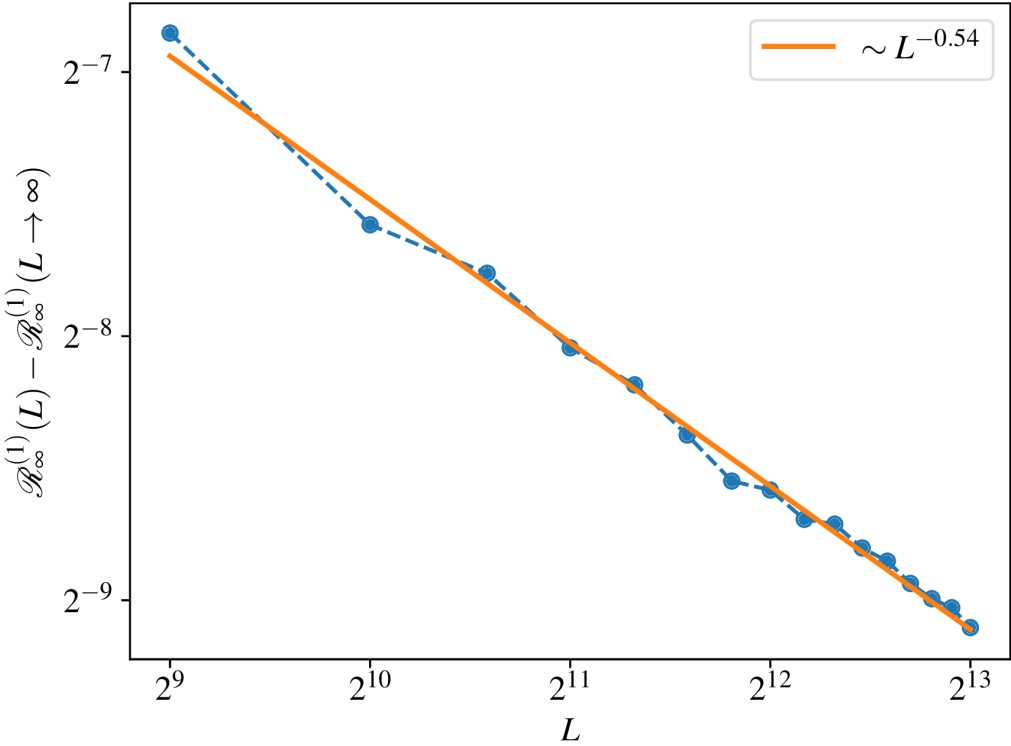

Figure 11: Subleading behavior of $\mathscr{R}_\infty^{(1)}$ obtained by subtracting out the leading behavior by extrapolating the data to $L \to \infty$.

two mutually incommensurate potentials in the continuum, with the Aubry-André lattice only a limiting case [20,21]. Thus, with both incommensurate potentials and periodic drives available in state of the art cold atom experiments, our work opens up a new avenue towards realisation of wavefunction multifractality. No fine-tuning is required, with multifractality exhibited by a *finite fraction* of the Floquet eigenstates for a range of driving strengths.

Avenues for future research are evident. For instance, what are the precise properties of the sub-diffusion exhibited by the Floquet multifractal states, and are the observed nontrivial exponents universal? Quite broadly, a key question for Floquet multifractality is the role of dimensionality, known to be a central ingredient for the physics of Anderson localisation. More narrowly, localisation in 'infinite-dimensional' hierarchical systems, where multifractality is ubiquitous, is often considered as toy model for many-body localisation [46] owing to the hierarchical nature of the Fock space. Hence, at a conceptual level, one may ask to what extent a Floquet system hosting multifractal states can be likened to models of many-body localisation, where interestingly the quasiperiodic nature of disorder can have a nontrivial influence on the localisation transition [47].

From a practical perspective, perhaps the most tantalising prospect is to ask how one can use robust multifractal states as basis for the realisation of other interesting phenomena, the possibilities of which are already hinted at by their potential role in enhancing the superconducting transition temperature in a quasi-1D superconductor via algebraic spatial correlations of multifractal states [48].

Note: During the consideration of the manuscript we have become aware of the other work [43] where the similar effective random matrix model (called preferred basis Levy matrix

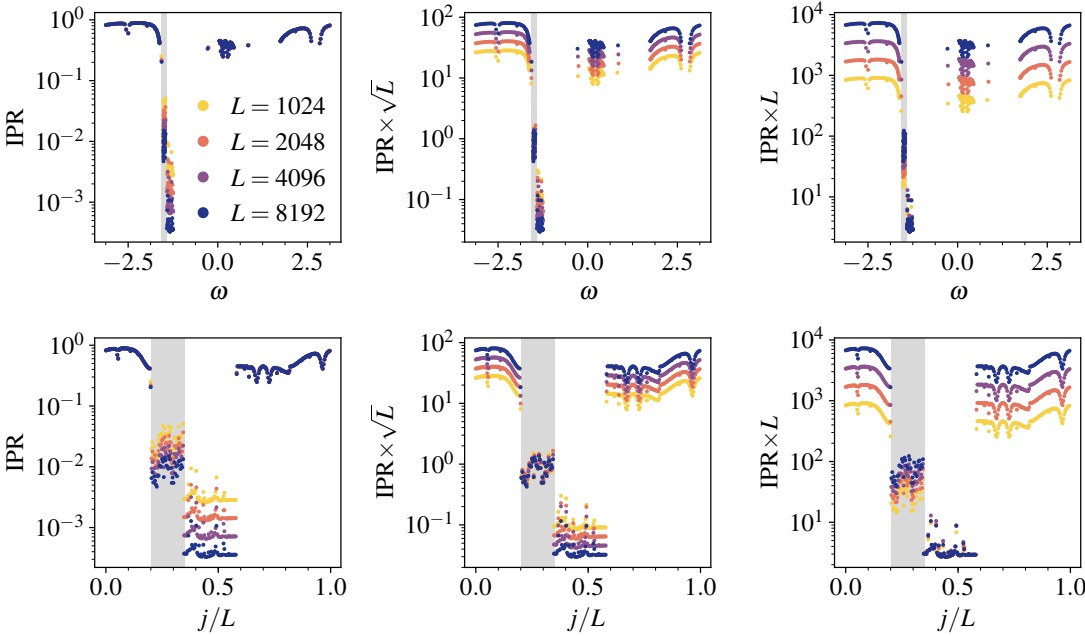

Figure 12: IPRs as a function of quasienergies. The top row panels show the IPRs unscaled, scaled with $\sqrt{L}$ and $L$, respectively as a function of quasienergy $\omega$. The gray shaded windows denotes the multifractal states, IPRs of which show reasonable collapse for various $L$ when scaled with $\sqrt{L}$. To clarify that there are a macroscopic number of multifractal states in the narrow quasienergy band, in the bottom panel we plot the same data but as function of quasienergy index.

ensemble) was obtained for a completely different system. Our results are consistent with the ones presented in [43] at $\gamma = 1.45 \pm 0.04$.

## Acknowledgements

The authors would like to thank G. De Tomasi, F. Evers, V.E. Kravtsov, A. Lazarides and A. Scardicchio for useful discussions and comments.

## A  Inverse participation ratios as function of quasienergy

In this appendix, we study the IPRs of the Floquet eigenstates as a function of their quasienergies. Since the quasienergy spectrum varies across disorder realisations, over many realisations, we bin the Floquet eigenstates within windows in quasienergy and average the IPR of the states within the windows. The results are shown in Fig. 12 where, in the top row the (scaled) IPRs are plotted as a function of the quasienergies $\omega$. The delocalised and multifractal states appear in a narrow separate bands of quasienergies. This is simply an artefact of the narrow bandwidth of the delocalised states in the undriven Hamiltonian (see Fig. 1) and that the multifractal states are born out of the delocalised states. To clarify this explicitly, in the bottom row in Fig. 12, we plot the (scaled) IPRs as function of bin labels ($j$) arranged in increasing order of quasienergy from $-\pi$ to $\pi$.

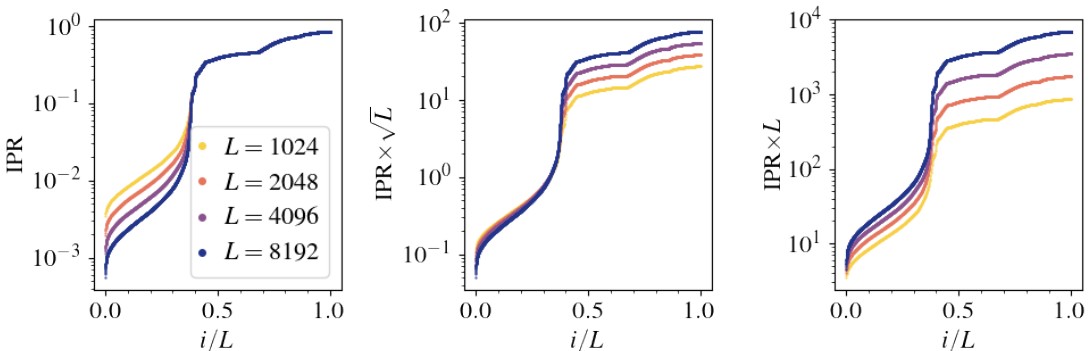

Figure 13: IPRs at lower driving frequency. The (scaled) IPRs similar to Fig. 2 for a much lower value of driving frequency $\Omega = 2.5\pi J$. The persistence of the multifractal states shows that no fine-tuning is needed in the driving parameters.

## B Robustness of Floquet multifractality to driving frequency

In order to show that the Floquet multifractality is robust to the driving frequency ($\Omega$) and $\Omega$ does not need to be fine tuned to close to but less than the bandwidth of the undriven Hamiltonian's spectrum, in this section, we show numerical evidence for the persistence of multifractality at lower frequencies as well. Recalling that the bandwidth of the undriven spectrum was approximately $2.76\pi J$, here we choose $\Omega = 2.5\pi J$ and in fact see an enhancement in the fraction of multifractal states in the Floquet spectrum. The results are shown in Fig. 13 in a similar fashion as Fig. 2. Note that there are almost no states whose IPRs scale as $1/L$. This is due to the fact that the lower frequency of the driving forces all the delocalised states in the narrow band at the top of the undriven spectrum (Fig. 1) to participate in hybridisations. On the other hand, the collapse of the IPRs when scale with $\sqrt{L}$ is worse. This might be attributed to higher order resonances for which the perturbative treatment based on the two-leg Shirley ladder is insufficient and its detailed analysis constitutes the topic for a future work.

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
