# Peer review of "Multifractality without fine-tuning in a Floquet quasiperiodic chain"

_SciPost Physics, doi:SciPost Phys. 4, 025 (2018)_

## Round 2 · Referee Report · Anonymous (Referee 1) · 2018-1-9

Strengths

The main idea of the paper is very interesting and should be published.

Weaknesses

The presentation can be greatly improved, and an important issue needs be clarified.

Report

1) Recommendations to improve the presentation :

1a) The present version of the manuscript contains one main text and 7 Appendices (from A to G) and 12 figures, so that the reading is extremely difficult : to understand a given statement mentioned in the text, one needs to go to the corresponding Appendix on another page, and to find the corresponding figures on still another page... (The three last Figures 10,11,12 are even plotted after the list of references !).

As a consequence, to facilitate the reading, I would recommend to include most of the material of the Appendices within the main text in order to obtain a logical presentation that can be followed linearly.

1b) Another difficulty in the reading comes from the mixing between exact and approximate statements almost everywhere, in particular in the discussion of multifractal exponents, for instance :

1b-i) the statement $D_q \simeq 1/2$ (end of the paragraph after Eq 3)
can only be valid in a certain region of the index $q \geq q_c $ that should be mentioned (since it cannot be true near $q=0$). Then in the caption of Fig 2 the authors give $\tau_2 \simeq 0.55$ while after Eq 5, one reads $D_q = 2(1-\gamma_1) \simeq 0.44$. And in figures 2 and 7 , the authors display the rescaling $\sqrt{L} I_2$ based on the value $1/2$.

1b-ii) the exponent beta \simeq 0.72 first appears in section 4 of the main text to characterize the sub-diffusive dynamics; then it appears after Eq 25 to characterize M_0 =L^{-\beta} with the value beta=0.9 \pm 0.1. Then in Eq 30, the fit value is beta=0.77, and three lines after Eq 30, the authors give again the value beta=0.9 \pm 1.

To avoid confusion between these different statements, I would recommend to clearly separate the definitions of the exponents and the exact relations between them, from the various numerical estimates.

1c) Equations : many equations are written inside the text, while they would be clearer if written as proper equations. Eq 21 should be written on a single lign. Eqs (31) and (32) seem to contain too big empty spaces ?

2) Important issue that needs to be clarified :

The multifractal spectrum given in Eq. 5 corresponds to the GAUSSIAN generalized Rosenzweig-Porter model of Ref [9], as discussed also in the various recent papers that the authors might wish to consider :

[R1] D. Facoetti, P. Vivo and G. Biroli, arxiv:1607.05942.

[R2] C. Monthus, arxiv:1609.01121

[R3] K. Truong and A. Ossipov, arxix:1609.03467.

[R4] B. L. Altshuler, L. B. Ioffe, V. E. Kravtsov, arxiv:1610.00758.

Here, the authors stress that their distribution of M is NOT GAUSSIAN (text around Eq 4 and Fig 4), and the form given in Eq. 26 actually corresponds to the Levy power-law tail at infinity 1/m^{1+\mu} of Levy index $\mu=2a-1 \simeq 1.2$ for the fit value $a \simeq 1.1$ given after Eq 26. As a consequence, the moments of $M$ alone already display the multifractal behaviors given in Eq 29, in contrast with the Gaussian case, so that the result of Eq 5 concerning the Gaussian case cannot be valid for the Levy case. The multifractal properties of some LEVY generalized Rosenzweig-Porter model have been actually discussed in the ref [R2] mentioned above. So the authors should clarify whether their effective matrix model with the distribution of Eq 26 has the same multifractal properties as the Levy model analyzed in the ref [R2] or not , and what are the consequences for their multifractal spectrum. (In particular, the derivation of Eq 24=Eq 5 in Appendix D based on the approach of Ref 9 is OK for the Gaussian case, but should be modified for the Levy case; the other approaches discussed in the refs R1-R2-R3-R4 might be more appropriate for the generalization to the Levy case).

In conclusion, I think that these clarifications on the precise effective matrix model and its multifractal properties are very important since the Floquet quasi periodic chain proposed by the authors gives a very nice realization of some generalized Rosenzweig-Porter model, while up to now, these models were only considered as 'toy models where multifractal spectra can be exactly computed'. It is thus interesting to understand precisely which variant of these models is realized in this Floquet framework, and what are its multifractal properties.

Requested changes

see the above report

---

## Round 2 · Referee Report · Anonymous (Referee 2) · 2018-2-22

Strengths

An original analysis of a simple model that leads to interesting results.

Weaknesses

Some statements in the abstract/intro are not explicitly demonstrated in the main text.

The presentation/layout is not as clear as it could be in places.

Report

This is an interesting paper. It addresses a theoretical question that arises naturally in the context of Floquet systems -- the effect of hybridising localised and extended states through the Floquet drive -- and finds the very interesting result that this leads to eigenstates with multifractal character.

The abstract refers to this as "an entire band of multifractal wavefunctions" and, correctly, emphasises that the multifractality is of interest because it "does not require any fine-tuning of the model parameters".

I do not doubt that these two quoted statements are correct. However, unless I have missed it in the paper, I do not see any direct evidence given for either statement.

To interpret this as "an entire band of multifractal wavefunctions" would surely require the reader to be given some sense of the (Floquet) energy spectrum of the multifractal states, as compared to the localised and delocalised ones. I do not see this is the paper, as the relevant graphs appear to sort the eigenstates by their IPR and not their quasienergy. Are the IPR and the quasienergy closely correlated? In what sense do the multifractal states form a band?

The results showing multifractal behaviour appear to be only for one value of the drive frequency: \Omega = 2.74\pi J. The authors argue why it needs to be close to the bandwidth, and in Appendix A show it should not be larger. But, unless I missed it, there is no statement about how robust the behaviour is to variations in \Omega about this value. Showing results even for one more value of \Omega would be helpful.

In terms of presentation, I did have trouble following some of the ideas, as I felt I needed to flick back and forward to appendices to understand. (The reference to Appendix F is particularly irritating, as one then needs to turn more pages to find the associated Fig 11.)

Requested changes

1) Please provide direct evidence justifying/explaining the statements ("an entire band of multifractal wavefunctions" and "does not require any fine-tuning of the model parameters") which are made in the abstract.

2) Please reduce need to consult appendices where possible (move what is important to the main text), and try to keep figures close to where they are referenced.

3) There are several typographical errors to correct:

page 2: localisation lengths is the unifies these two contexts.

Page 3: generalisations of which is known to host multifractal eigenstates

Page 7: This visible in a plot of σ2(n)/n2,

Page 9: To extract τq shown in Fig. 2(d) of the, we do a linear fit o

Fig 6 caption: The black dashed line denotes the mobility edge, states near which do not participate in any significant way.

Fig 7 caption: where as the columns correspond

---

## Round 3 · Referee Report · Anonymous (Referee 1) · 2018-4-5

Report

The authors have taken into account all the comments of the previous reports,
so I strongly recommend the publication of this revised version,

---

## Round 3 · Referee Report · Anonymous (Referee 2) · 2018-4-8

Report

The authors have responded in detail to all comments made, including the addition of new data to clarify one question. I recommend publication of the paper in its present form.

---

## Round 3 · Author Response

Dear Editor,

We thank your editorial recommendation and also the referees for considering our work suitable for Scipost, and for various useful suggestions. In the resubmitted version, we have taken all the suggestions and questions of the referees. A common suggestion of both the referees was to incorporate much of the appendices in the main text so that the paper is easier to read. In view of this, we have completely restructured the paper so that the main text, now split up into sections, is completely self-contained and appendices are used only to show additional numerical data.

Below, we address the specific questions of the referees in detail.

We hope that the concerns of the referees have been addressed satisfactorily, such that the work can now be published in Scipost.

reply to Report #1

  1. Regarding improving the presentation:

    (a) We have completely restructured the main text of the manuscript to make it as self-contained and linear as possible and have used the appendices to present only some additional numerical data.

    (b) We have tried to be very clear and explicit regarding what values are exactly obtained from the numerical results and which of them are approximate values obtained either from the fits and or from the perturbation theory. (i) The quoted values of $D_q$ are valid for $q\gtrsim 1$; we mention this clearly now. (ii) The multifractal exponents show a small spread over all the multifractal states in the spectrum. The value 0.55 is obtained after averaging over all the multifractal states. The scaling with $\sqrt{L}$ based on the value 0.5 is shown to facilitate the demonstration of the non-trivial scaling of the IPRs with system size. The value of 0.44 is obtained from the proposed random matrix model for the Floquet multifractal states which is further analysed perturbatively. We have been explicit in distinguishing these now. (iii) We apologise for reusing the notation $\beta$ in two different contexts. In the previous version of the manuscript, the exponent $\beta$ in the section on wavepacket dynamics is not the same as the one used in the analysis of the distributions of matrix elements. To clear the confusion, we now use the symbol $\nu$ for the latter.

    (c) We have taken care of the formatting issues of equations raised by the referee.

  2. Regarding the question about Levy matrices:

    We first thank the referee for bringing to our attention the references on generalised Rosenzweig-Porter and the one on the multifractal states in Levy random matrices. We have cited them appropriately. The Referee is correct in saying that the results and analysis of the Rosenzweig-Porter random matrices do not immediately carry over to our case due to the power-law tails of the distributions of the matrix elements. However, the perturbative analysis we do is robust to that as can be shown by a one-parameter rescaling of the distribution as a function of the system size. We have modified the text in Secs. 4.2 and 4.3 to explicitly state the assumptions (4.2) and their justifications (4.3) and substantiate them with the numerical analysis of the distribution. We would like to mention that the exponents obtained from the rescaling of the distributions are not very accurate, however, there are a couple of points to be noted here. Firstly, the random matrix model proposed for the Floquet multifractality already has a few assumptions built in which could lead to the errors, and secondly, as can be seen in Fig. 8(b) of the current version, it is difficult to get very reliable statistics for the tails of the distributions. On the other hand, the encouraging feature is that the scaling of the mean and typical values of the distributions with system sizes are much more accurate when compared to the exact numerical results. To be consistent, we also cite another realisation of the generalised Rosenzweig-Porter ensemble with a Levy-type distribution of the hopping elements dubbed as “preferred basis Levy matrix ensemble” in V. N. Smelyansky et al. arXiv:1802.09542. In this example, multifractal states also have the same properties as in the generalised Rosenzweig-Porter ensemble. So this validates our analytical perturbative calculations.

reply to Report #2

  1. The referee raises a question about the quasienergy structure of the multifractal states and asks for a justification of the statement "an entire band of multifractal wavefunctions". As is the case naturally in disordered/quasiperiodic systems, the energy spectrum varies across disorder realisations and hence sorting the Floquet eigenstates with their IPRs was rather convenient for the purpose of scaling the IPRs and higher moments. However, to address the referee's concern, we have now shown additional data in Appendix A where we have presented the IPRs and their scalings with system sizes as a function of the quasienergy. The results show that the multifractal states indeed appear in a contiguous band of quasienergies thus justifying our statement. To demonstrate this, we bin the Floquet eigenstates over many realisations within windows in quasienergy and average the IPR of the states within the windows.

  2. The referee asks about the robustness of the Floquet multifractality to the driving frequency as we had chosen to show data for only one frequency which was rather close to the bandwidth of the undriven system. To address this issue, in Appendix B, we have shown IPRs and their scalings for a much lower value of the driving frequency. We find that not only is the multifractality persistent, but its fraction is enhanced due to more delocalised and localised states taking part in the hybridisations.

  3. The referee suggests reorganising the manuscript to minimise the need for appendices and also points out typographical errors. We have now completely restructured the manuscript by incorporating much of the appendices in the main text so that the paper is easier to read, and have also taken care of the typographical errors.

Additional corrections

1) We have slightly modified the derivation in Eqs. (16-19) and (25) to confirm the applicability of our perturbative approach to the case of Levy distribution of hopping elements.

2) We have replaced the notation $\gamma_2$ by $\gamma/2$ to be consistent with the previous papers on the generalised Rosenzweig-Porter ensemble.

3) To clarify the estimation of the exponent $\gamma=2-D$ we have merged Figs. 4 and 10 and modified the panel showing the scaling of the moments with L focusing on the typical and mean instead of higher moments.

4) We have added the references [35-40] and [43] considering the generalised Rosenzweig-Porter ensemble.

---

## Round 3 · List of Changes

1) We have completely restructured the main text of the manuscript to make it as self-contained and linear as possible and have used the appendices to present only some additional numerical data. 2) We have slightly modified the derivation in Eqs. (16-19) and (25) to confirm the applicability of our perturbative approach to the case of Levy distribution of hopping elements. 3) To clarify the estimation of the exponent \gamma=2-D we have merged Figs. 4 and 10 and modified the panel showing the scaling of the moments with L focusing on the typical and mean instead of higher moments. 4) We have added additional data to show the robustness of the Floquet multifractality to variations in the drive frequency. 5) We have shown additional data where the IPRs are shown as a function of quasienergy to justify the term "band of multifractal states".

---

## Editorial Decision

published